# Rhizosphere Colonization Determinants by Plant Growth-Promoting Rhizobacteria (PGPR)

**DOI:** 10.3390/biology10060475

**Published:** 2021-05-27

**Authors:** Gustavo Santoyo, Carlos Alberto Urtis-Flores, Pedro Damián Loeza-Lara, Ma. del Carmen Orozco-Mosqueda, Bernard R. Glick

**Affiliations:** 1Instituto de Investigaciones Químico-Biológicas, Universidad Michoacana de San Nicolás de Hidalgo, Morelia 58030, Mexico; 1316102a@umich.mx; 2Licenciatura en Genómica Alimentaria, Universidad de La Ciénega del Estado de Michoacán de Ocampo, Sahuayo 59103, Mexico; pdloeza@ucemich.edu.mx; 3Facultad de Agrobiología “Presidente Juárez”, Universidad Michoacana de San Nicolás de Hidalgo, Melchor Ocampo, Uruapan 60170, Mexico; carmen.orozco@umich.mx; 4Department of Biology, University of Waterloo, Waterloo, ON N2L 3G1, Canada; glick@uwaterloo.ca

**Keywords:** rhizosphere, biocontrol, bioinoculants, sustainable agriculture

## Abstract

**Simple Summary:**

Plant growth-promoting rhizobacteria (PGPR) are an eco-friendly alternative to the use of chemicals in agricultural production and crop protection. However, the efficacy of PGPR as bioinoculants can be diminished by a low capacity to colonize spaces in the rhizosphere. In this work, we review pioneering and recent developments on several important functions that rhizobacteria exhibit in order to compete, colonize, and establish themselves in the plant rhizosphere. Therefore, the use of highly competitive strains in open field trials should be a priority, in order to have consistent and better results in agricultural production activities.

**Abstract:**

The application of plant growth-promoting rhizobacteria (PGPR) in the field has been hampered by a number of gaps in the knowledge of the mechanisms that improve plant growth, health, and production. These gaps include (i) the ability of PGPR to colonize the rhizosphere of plants and (ii) the ability of bacterial strains to thrive under different environmental conditions. In this review, different strategies of PGPR to colonize the rhizosphere of host plants are summarized and the advantages of having highly competitive strains are discussed. Some mechanisms exhibited by PGPR to colonize the rhizosphere include recognition of chemical signals and nutrients from root exudates, antioxidant activities, biofilm production, bacterial motility, as well as efficient evasion and suppression of the plant immune system. Moreover, many PGPR contain secretion systems and produce antimicrobial compounds, such as antibiotics, volatile organic compounds, and lytic enzymes that enable them to restrict the growth of potentially phytopathogenic microorganisms. Finally, the ability of PGPR to compete and successfully colonize the rhizosphere should be considered in the development and application of bioinoculants.

## 1. Introduction

Agriculture is essential for the food security of humans and animals that live on the planet [1]. It has been predicted that by 2050, the human population could reach 8 billion, which will present a significant challenge for agricultural systems to produce enough food to feed this global population, especially given the fact that there are a wide range of biotic and abiotic factors that have a significant negative impact on agricultural productivity [2]. Among the limiting biotic factors, there are a multiplicity of pathogens such as bacteria, fungi, viruses, insects and nematodes. The successful management of these pests is essential to avoid losses during production [3]. Currently, pest management is carried out mainly through the use of pesticides and agrochemicals, which do not completely solve the problems caused by the various phytopathogens, as they can generate negative effects, such as health problems, loss of ecological diversity, and the bioaccumulation of toxic substances [4].

The abiotic factors that are inhibitory to plants include extreme variations in temperature, salinity, soil contamination, flooding and drought [5]. In addition, an excess of chemical fertilizers in the soil can also decrease soil fertility, inhibit the beneficial micro- and macrobiota, and contaminate the aquifers [6,7]. One of the approaches that may be used to solve some of the current problems of agriculture is the application of beneficial microorganisms that are naturally associated with plants, which are ecological alternatives that do not have secondary effects on the environment, human health, and animals [8,9,10]. This biota is known as the plant microbiome, and different positive effects are attributed to it, such as the stimulation of plant growth and protection against pathogens [11,12]. There are multiple mechanisms that this microbiome can utilize to benefit the plant, including the production and modulation of phytohormones, solubilization and increasing the bioavailability of essential nutrients, production of antibiotics, synthesis of volatile compounds and secondary metabolites, and improvement of the physical and chemical properties of the soil [13,14].

Within the plant microbiome, plant growth-promoting rhizobacteria (PGPR) can colonize and proliferate within the rhizosphere environment [15,16]. The rhizosphere is defined as the area of the soil immediately surrounding the root, which is directly influenced by the plant root exudates [17]. Root exudates include a range of organic acids, amino acids, sugars and other small molecules exuded by the plant roots that act as strong chemo-attractants of the soil microbiota. Thus, depending on the plant species, or even the variety, the roots can produce important differences in the chemical composition of the exudates, which attract a particular microbial diversity [18,19]. In turn, the soil microbiota can detect particular chemical signals that allow it to efficiently colonize specific rhizospheres and plant roots. A widely known example of specific plant–bacteria communication is that of legume-rhizobia [20,21]. Legume plants produce flavonoid compounds (among other molecules) that are secreted through the roots, which are recognized by rhizobia, which in turn produce nodulation factors (Nod factors) that induce the formation of nodules on the roots. Once the nodules are formed and colonized by rhizobia, they can fix atmospheric nitrogen and make it available to the plant in the form of ammonia, thus significantly benefiting plant growth and nutrition [22].

However, not only does the plant exude compounds that attract a particular microbiome, but there are also mechanisms present in PGPRs that allow them to recognize plant molecules, acquire nutrients, occupy spaces, and either directly or indirectly inhibit other microbial species [14], in order to survive and colonize the rhizosphere. An excellent review of the molecular determinants of colonization of the rhizosphere by *Pseudomonas* spp. [23] laid the foundations of the importance of deciphering the mechanisms of rhizosphere colonization, as well as highlighting areas of opportunity for research. Therefore, understanding rhizosphere colonization mechanisms by PGPR is essential for generating inoculants able to compete and efficiently colonize the rhizosphere of plant crops, and having a great impact on crop production and more consistent results. (Figure 1).

## 2. Root Exudates as a Selective Factor in the Microbiome

The energy emitted by the sun is captured and used for the synthesis of compounds by plants through the process of photosynthesis, which is essential for the maintenance of ecosystems and trophic chains. The photosynthates generated by plants are secreted by the roots (rhizodeposition) [24], which makes the rhizosphere an area very rich in nutrients, and therefore, one of the micro-zones of the soil with the greatest diversity and ecological richness [19,25]. Root exudates comprise up to 21% of the carbon fixed by plants and consist of various organic compounds such as amino acids, fatty acids, nucleotides, organic acids, phenolics, plant growth regulators, sugars, sterols, and vitamins [26]. Thus, the rhizosphere, in addition to being rich in nutrients, is an ecosystem where there is high competition for the exuded nutrients between the different microorganisms and organisms that inhabit it.

A two-step microbiome acquisition model has been proposed, which suggests that the first filter for these microorganisms is to go through various processes of cell recognition, at the level of the membrane and cell wall characteristics, as well as a substrate-dependent movement from the soil, which contains different compounds secreted by the roots. In the second step, this subpopulation of microbial members goes through a second selection process, imposed by the genetic characteristics of the host, and that are specific for the selection of certain groups of microorganisms [25,27]. Thus, the genotypic properties of the plant favor and are associated with certain microbiomes. The authors of [28] demonstrated that the different root exudates of plants positively or negatively modulated individual fungal phylotypes, exerting strong selective pressure to structure the community that successfully colonizes the rhizosphere of a particular plant. The authors came to this conclusion by quantitatively and qualitatively evaluating the root exudates of *Arabidopsis thaliana* and *Medicago truncatula* plants and their impact on the composition of resident and non-resident soil fungal communities [28].

In a recent study, it was observed that the secretion of 2,4-dihydroxy-7-methoxy-1,4-benzoxazine-3-one (DIMBOA) from maize species (*Zea mays* L.) inhibits the development of phytopathogens but allows the proliferation of beneficial bacteria [29]. In addition, other plant species such as tomato (*Solanum lycopersicum*) can rhizo-deposit malic acid, which functions as a substrate and attractant for beneficial bacteria such as *Bacillus subtilis*, which, in turn, secretes compounds such as bacilomycin that inhibit the development of potential phytopathogens [19]. This secretion of compounds by the plant has distinctive patterns throughout the root, generating microenvironments with different physicochemical properties, each resulting in a microbial community with a particular biodiversity and structure [30].

Therefore, the rhizosphere of a plant is an area where there may be selective pressure toward attracting certain microbial communities from the bulk soil. A recent study of microbial communities in the rhizosphere of 19 plant species used molecular tools such as 16S rRNA gene sequencing to demonstrate that each analyzed plant species could “select“ between 18 and just over 100 operational taxonomic units or OTUs of more than 1000 found. The authors concluded that this minority of species may have a significant impact on plant growth, as well as on plant–bacteria interactions [31].

Considering the importance of root exudates as chemo-attractants of beneficial bacteria (among other neutral and pathogenic organisms) in the rhizosphere, [32] used a co-inoculation strategy that included root exudates of corn plants and *Azospirillum brasilense* strain Ab-V5 together with corn seeds. The results showed that the biomass of the corn seedlings supplemented with the root exudates plus the *A. brasilense* strain increased the root area by 50% and the number of bacteria per plant by 19% compared to treatments where only the bacteria were inoculated onto the seeds. The authors conclude that the compounds exuded by corn seedlings can increase the colonization of the corn rhizosphere by *A. brasilense* Ab-V5 and may be a strategy that could make bioinoculant formulations more efficient.

## 3. Acquisition and Nutrients Solubilization

As previously mentioned, the rhizosphere is a microenvironment that contains a large quantity and diversity of nutrients, which must be acquired by the PGPR to proliferate, move, compete, and colonize spaces on the root. For this reason, some species, including Proteobacteria (*Pseudomonas*, *Burkholderia*, *Rhizobium*, etc.), Firmicutes (*Bacillus*, *Peanibacillus*, *Neobacillus*, etc.), and Actinobacteria (*Arthrobacter*, *Actinomyces*, *Micrococcus*, *and Streptomyces*), are among the most common inhabitants of the rhizosphere [33,34]. Therefore, a characteristic that these bacterial groups have is the ability to acquire and utilize these nutrients through various mechanisms, some of which are described below.

### 3.1. Siderophores and the Acquisition of Iron

Siderophores are low-molecular-weight secondary metabolites produced by PGPR (and other microbes) in iron deficiency states to bind iron and supply it to the bacterial cells [35,36]. In the rhizosphere, crops associated with siderophore-producing PGPR may obtain iron through microbially produced siderophores [37]. These molecules have the ability to capture metal ions with a much greater affinity for Fe (III) than Fe (II). Depending on the functional group that acts as the sequestrant, they can be classified into catecholates, hydroximates, and hydroxycarboxylates [38]. There are more than 500 biomolecules that are classified as siderophores; therefore, different genes and regulators are involved in their biosynthesis, transport, and re-importation into the cell [37].

These iron-chelating compounds, which are widely produced by PGPR species, confer an advantage over those microorganisms unable of producing them. Iron deficiency can lead to severe biological inhibition for organisms by depriving them of this element because it is essential in cellular processes such as DNA synthesis, respiration, and free-radical detoxification [39].

Siderophores are produced by a wide range of bacterial, fungal, and plant species [40]. Likewise, the siderophores produced by PGPR have been attributed to various functions in the rhizosphere. In addition to conferring an advantage to take iron into the rhizosphere, particularly under limiting conditions, siderophores may also inhibit the growth of pathogens that could potentially cause damage to the plant [41]. PGPR that produce siderophores in rhizospheres with little iron can bind available iron and make it less available to pathogens, indirectly promoting plant growth. Such is the case of the siderophores produced by the bacterium *Burkholderia cenocepacia* strain XXVI, which produces iron-chelating siderophores with biocontrol activity against the fungal pathogen *Colletotrichum lindemutianum* ATCC MYA 456 [42]. In addition, it has long been known that siderophores produced by *Pseudomonas* spp. are important in promoting plant growth [43,44]. Therefore, the production of these compounds is an important factor in the colonization of spaces in the rhizosphere under iron-limiting conditions and exhibits plant growth-promoting activities.

### 3.2. Phosphate Solubilization

Phosphorus is one of the most important elements for agricultural plant production. It can be found in two forms: organic phosphorus (generally 30 to 50% of the total), which is mostly in the form of inositol hexaphosphate, or phytate, which is the principal storage form of phosphorus in plants that can be degraded by bacteria or fungi; and inorganic phosphorus, which usually forms insoluble mineral compounds with calcium, aluminum, or manganese [45,46]. Most soils contain a large amount of phosphorus [45]. On the other hand, most of this phosphorus is insoluble and is not available to support plant growth. The amount of soluble phosphorus in most soils is around 1 mg/kg of soil, which is insufficient to support plant growth [45].

The distribution of these forms of phosphorus in soils depends on factors such as microbial activity, pH, soil type, and the availability of organic matter. In addition to phytate, which cannot be used by plants unless it is first enzymatically broken down, the phosphorus available to plants is in the form of orthophosphates and soluble inorganic forms such as monobasic (H_2_PO_4_^−^) and dibasic (HPO_4_^2−^) ions. As the availability of this element is low, plants and microorganisms compete for it through precipitation, solubilization, absorption, and desorption processes. Organic phosphorus can be mineralized as a by-product of the mineralization of soil organic matter or through the action of specific enzymes that are regulated by the demand for this nutrient. The main mechanism involved in the solubilization of inorganic phosphate involves the synthesis and exudation of organic acids by soil bacteria. These organic acids typically originate from glucose oxidation [4]. 

Although there is no obvious relationship between phosphate solubilization and rhizospheric colonization capacities, PGPR with phosphate-solubilizing activities can survive in the rhizosphere either by direct phosphorus uptake or due to the bioavailability that promotes better development of the plant, including the root system. Here, it should be noted that mycorrhizal fungi, that are estimated to beneficially interact with the roots of >90% of all vascular plants, also sequester and solubilize phosphorus from the soil and provide it to plants [47].

### 3.3. Nitrification and Nitrogen Fixation

Nitrogen is one of the most important elements in plant synthesis, as it is a constituent of nucleic acids, peptides, organic acids, and fatty acids, which are essential for the structure and functioning of all living beings. Nitrogen can be captured and fixed in soils by diazotrophic bacteria, which are responsible for the fixation of atmospheric nitrogen into ammonia, which is the initial substrate for the nitrification process [48,49,50]. The nitrification process subsequently involves the transformation of ammonium to nitrate and is carried out by nitrifying bacteria, such as *Nitrosomonas* spp. or *Nitrobacter* spp. [51]. This process has two main steps. First, ammonium ion (NH_4_^+^) is transformed into NH_2_OH by the action of the enzyme ammonium monooxygenase that catalyzes the oxidation process, requiring two electrons for the reduction of an O_2_ atom in H_2_O, and NH_2_OH is converted to NO_2_ by the action of the hydroxylamine oxidoreductase enzyme, which is carried out by a group of ammonium-oxidizing bacteria. The second step involves the transformation of NO_2_ into NO_3_ through the catalysis generated by the enzyme nitrite oxidoreductase secreted by nitrite oxidizing bacteria (NO^2−^), which are chemolithoautotrophs; that is, they use the chemical energy of nitrification to fix CO_2_ [48].

There are also bacteria collectively called rhizobia (*Rhizobium*, *Mesorhizobium*, and *Bradirhizobium*) that are capable of associating in a symbiosis (highly regulated and specific) with the roots of legume plants [22,52]. As discussed above, legumes produce flavonoids, which are recognized by rhizobia, which, in turn, induce the production of molecules called nodulation factors. These Nod factors induce a morphological transformation in the root to form a globular structure called a “nodule”. In nodules, rhizobia can fix atmospheric nitrogen [53]. The fixation of atmospheric nitrogen by microorganisms is also evaluated by identifying the *nifH* gene, which encodes the Fe protein of the nitrogenase enzyme complex [21]. Thus, nitrification and nitrogen fixation processes are important in the nitrogen cycle because they fulfill an important ecological function in ecosystems, including those of agricultural interest.

## 4. Antioxidant Activities

The rhizosphere is an environment that can be stressful for the microbiota that inhabits it, including the PGPR that are associated with plants, particularly those soils where there are various types of abiotic stresses, including salinity, drought, the presence of heavy metals, or extreme levels of alkaline or acidic pH [54]. This type of abiotic stress can, in turn, cause oxidative stress and consequently damage biomolecules in bacterial cells. Therefore, PGPR must contain mechanisms to protect against various types of environmental stress.

Oxidative stress is defined as an imbalance between the proportion of genobiotic oxidants as allogeneic and molecules with antioxidant properties in a biological system. These oxidizing agents are usually compounds derived from oxygen or nitrogen that have chemically reactive forms due to unpaired electrons in their surface energy layers, which are called reactive oxygen species (ROS). These compounds can have various chemical structures, forming anions, peroxides, superoxides, and radicals such as hydroxyl, alkoxy, peroxyl, nitrogen dioxide, lipid hydroperoxide, and hyperchlorites [55]. PGPR mechanisms can mitigate the harmful effects caused by oxidative stress, using both non-enzymatic and enzymatic methods, particularly under saline or drought stress [56]. In the case of non-enzymatic methods, they are aimed at reducing exposure to ROS, such as migrating to spaces where there is less solar radiation, the production of pigments that absorb some of the harmful radiation, and the packaging of DNA with chromatin and proteins to provide alternative sites for the attack of these reactive species. There are also non-enzymatic antioxidant compounds that in low concentrations prevent or delay the oxidation of oxidizable substrates such as alpha tocopherol or vitamin E, ascorbic acid or vitamin C, carotenoids, flavonoids, trehalose and reduced glutathione [57]. On the other hand, enzymatic methods make use of intelligently designed enzymes with the aim of not generating more reactive species; they can also transform these products into molecules with less harmful properties or locate them in cellular substructures that will later be degraded; these methods also allow the maintenance of ROS at appropriate physiological levels. Some enzymes that are included in this category are superoxide dismutases, catalases, glutathione peroxidases, glutathione sulfate transferases, quinone reductases, and peroxiredoxins [58]. 

The mechanisms of resistance to oxidative stress and detoxification of ROS are not only important to colonize rhizospheric environments, but also to the internal compartments of the plant. This was demonstrated by Alquéres et al. [59] through the construction of mutants in genes involved in ROS detoxification, such as glutathione reductase and superoxide dismutase in the N_2_-fixing endophyte *Gluconacetobacter diazotrophicus* PAL5, which plays an important role in the colonization of the endosphere of rice (*Oryza sativa* IR-42) seedlings. On the other hand, there is also evidence that antioxidant defense mechanisms tend to be more efficient in multicellular conditions, such as the production of biofilms by some bacteria [60].

## 5. Biofilm Production

Biofilms are extracellular matrices composed of exopolysaccharides, proteins, nucleic acids, lipids, and microorganisms embedded in them [61,62]. Lipopolysaccharides (LPS), especially the O-antigen, can play important roles in root tip colonization, as demonstrated by Dekkers et al. [63] in *Pseudomonas* sp. However, biofilm production is not a specific characteristic of a particular bacterial group. Bacteria of the genera *Bacillus*, *Pseudomonas*, *Vibrio*, *Staphylococcus*, and *Salmonella* are the genera most commonly used to study the process of biofilm production. Some of these organisms have a clinical interest, but this ability also allows those rhizospheric bacteria to adhere to the surface of plant roots [62]. Once attached to the plant roots, PGPR can more readily exert their beneficial mechanisms towards the plant; PGPR with good biofilm production increase their plant growth-promoting activities, including those situations where there is an environmental stress [64,65].

The formation of a biofilm matrix is carried out in stages. Initially, microorganisms adhere to a surface in what is known as primary adherence; these microorganisms have different cellular structures such as pili or flagella and enzymes called adhesins that facilitate this adherence. Motility can help microorganisms to counteract the hydrophobic forces that often repel them from surfaces. In the second stage, the microorganisms that managed to adhere successfully begin to divide, spread around the initial site, and form microcolonies. The next stage begins with the secretion by the microorganisms of different exopolysaccharides including alginates, celluloses, n-acetylglucosamines, and galactose. Finally, the microcolonies embedded in the exopolymer matrix begin to free themselves from the matrix and may repeat the process at a different site [61]. A biofilm often confers increased capacity to the integrated microorganisms. The genus *Pseudomonas* is one of the most well-studied biofilm-producing microorganisms. 

The phenomenon of *quorum sensing* has often been studied along with biofilm formation. This process coordinates and regulates the expression of genes and compounds at the population level through a specific chemical language based on molecules derived from N-acyl homoserine lactones, thus improving the efficiency of the action of secreted secondary metabolites that can exert beneficial effects on plants or impose selection on other microorganisms that compete for a niche [66]. *Quorum sensing* enables groups of bacteria to behave in a coordinate manner; once cell densities attain a critical level, *quorum sensing* enables bacteria to switch on different sets of genes, facilitating concerted interactions between the cells. The rhizosphere is a site of genetic transfer between bacteria, which increases their functional profile and the permanence of resistance genes through generations and species. Biofilms produced by PGPR also confers protection to plants subjected to stress conditions, such as drought and hypersalinity, since its constituents may coordinately function as osmoprotectors [67]. 

The direct role of biofilm formation and colonization capacity was confirmed by Meneses et al. [68]. These authors identified a gum gene cluster in the genome of the nitrogen-fixing bacterium *Gluconacetobacter diazotrophicus* PAL5, which was predicted to be responsible for the first step in exopolysaccharide production (EPS). The *gum* PAL5 mutant showed normal growth and nitrogen (N_2_) fixation levels but did not produce EPS or biofilm when grown on different carbon sources. In addition, the gum mutant was unable to efficiently attach to the roots of rice plants or to colonize the internal tissues. Thus, biofilm production is a characteristic that allows PGPR to survive different types of environmental stress and, in turn, maintain high cellular levels attached to the roots of the plants, from where they can exert their beneficial interactions in the rhizosphere [69].

## 6. Volatile Organic Compounds

Volatile organic compounds (VOCs) are low-molecular-weight molecules that have a vapor pressure of 0.01 kPa or more at room temperature, usually contain fewer than 12 carbon atoms, and may be associated with other elements such as nitrogen, sulfur, bromine, oxygen, fluorine, and chlorine [70]. When these compounds are produced by organisms, they are called biogenic VOCs and have been shown to be important in different processes by which they promote plant development, the induction of systemic resistance (ISR), and chemical signaling in plants [71,72].

An example of these VOCs with beneficial activities in plants is produced by the rhizobacterium *Artrobacter agilis* strain UMCV2, which synthesizes the volatile dimethylhexadecylamine (DMHDA), involved in promoting the growth and development of *Medicago truncatula* seedlings, particularly under conditions of iron scarcity. DMHDA increased the chlorophyll content, biomass, and ferric reductase activities. In addition, DMHDA stimulated the roots of *M. truncatula* to exude protons that facilitate acidification of the rhizosphere, allowing the uptake of iron under limiting conditions and increasing the content of this element in plants treated with this bacterium [73]. In a later study, it was observed that strain UMCV2 has the ability to colonize the internal tissues of plants and survive as an endophyte [74,75].

There is also evidence that VOCs can stimulate the immune system of plants and have a beneficial impact on the rhizosphere. For example, *Bacillus subtilis* produces volatile 2,3-butanediol, a compound that induces growth and systemic responses in plants. When 2,3-butanediol was directly applied to plant roots, followed by exposure to the pathogen *Ralstonia solanacearum*, the expression of pathogenesis-related (PR) genes, including *CaPR2*, *CaSAR8.2*, and *CaPAL*, was observed. Moreover, plant roots exposed to 2,3-butanediol responded by increasing the production of root exudates. These results suggest that 2,3-butanediol triggers the secretion of root exudates that modulate the functioning of soil rhizosphere fungi and bacteria [76]. In addition, some VOCs function as chemical signaling molecules between domains; some bacteria can secrete VOCs with antimicrobial properties such as dimethylsulfite. This was demonstrated by Rojas-Solis et al. [77], who observed that two *Bacillus* strains produced volatile compounds with synergistic activities to promote tomato plant growth and antagonize potential pathogens such as *Botrytis cinerea*. One of the volatile compounds was dimethyl disulfide (DMDS), which, when added in its pure form, was able to inhibit the mycelial growth of *B. cinerea*. Not all of these compounds, such as DMDS, have as yet been evaluated for their specific role in rhizosphere colonization.

## 7. Production of Antimicrobial Compounds

The microorganisms associated with plants are under constant nutritional competition, which is why they have developed the synthesis of various antimicrobial compounds as a strategy to compete with other microorganisms for establishment in a specific niche [78]. Based on their antimicrobial effect, they have been classified as bactericidal to denote compounds that have a lethal effect on cells and bacteriostatic agents that temporarily inhibit the development of microorganisms [79]. These antimicrobial compounds have a fairly broad chemical nature, which allows them to act on different cellular targets and interfere with various processes of microorganisms, such as organic acids that modify the pH of the medium, thus imposing a selection on other microorganisms. Some plants that are under attack by phytopathogenic fungi recruit microorganisms to deal with this infection. For example, wheat plants that are inhibited by *Gaeumannomyces graminis* recruit *Pseudomonas* spp. that generate the compound 2,4 diacetylphloroglucinol, which has antifungal properties [34,80]. Moreover, the secretion system in rhizobacteria is important for the excretion of antibiotic protein molecules, in order to be competitive in the rhizosphere [81]. Secretion systems such as T6SS are widely present in Proteobacteria and were originally considered as virulence factors; however, recent studies have assigned a role in rhizosphere adaptation in *Pseudomonas* species to T6SS systems, as evidenced by the work of Durán et al. [82]. These authors showed that a double mutant in F1- and F3-T6SS gene clusters was severely impaired in persistence in the rhizosphere microbiome of tomato plants, suggesting that the mutant strains exhibited a decrease in bacterial antagonism in such soil microecosystems, and therefore, were less competitive.

### 7.1. Lytic Enzymes

The cell walls of fungi and oomycetes are composed of chitin, cellulase, and glucan, among other molecules. Therefore, they are the target of some lytic enzymes produced by PGPR, including β-1,3-glucanases, lipases, cellulases, and chitinases [83]. Such is the case of *Bacillus thuringiensis* UM96, a rhizospheric bacterium that produces chitinases against the pathogen that causes gray mold, *B. cinerea* [84,85]. Assigning a specific role for chitinases, if they cannot be purified, is difficult, since other lytic enzymes such as glucanases or cellulases can also act as antifungal compounds. Therefore, the authors used a specific chitinase inhibitor (allosamidin) and observed that the antifungal activity of strain UM96 was suppressed in supernatants supplemented with allosamidin at low concentrations (100 µM). Consistent with this result, strain UM96 did not show any other cell wall-degrading activities. Moreover, strain UM96 can interact synergistically with other rhizospheric strains such as *P. fluorescens* UM16, UM256, UM240, and UM270, and promote the growth of corn seedlings [86]. Other enzymes such as cellulases, which are specialized in the degradation of cellulose and other cell wall polymers, play an important role not only in antagonism towards pathogens but also in the ability to colonize the endosphere of plants [87]. This is due to the fact that the cell walls of plants contain some of these target molecules and are susceptible to being attacked. However, PGPR with cellulolytic properties do not visibly cause damage to plant tissues.

### 7.2. Antibiotics

The production of antibiotics by PGPR gives them a competitive advantage in the rhizosphere, by eliminating or stopping the growth of many bacterial and fungal pathogens, for which the production of these compounds has been strongly associated with the ability to colonize the rhizosphere [88,89]. For example, Bais et al. [90] showed that the production of biofilms in several strains of *Bacillus subtilis* was essential for colonization and establishment in the rhizosphere of *A. thaliana* plants, and once the bacterium was established, it produced an antibiotic with protective action against potential infections caused by pathogenic bacteria such as *P. syringae* pv tomato DC3000.

Another study that showed the action of three antimicrobial compounds, surfactin, fengycin, and iturin A, with a relevant role in the suppression of powdery mildew in cucurbits caused by the fungal phytopathogen *Podosphaera fusca* [89]. The authors evaluated the supernatants of four *Bacillus* strains (UMAF6614, UMAF6616, UMAF6639, and UMAF8561) that contained the antibiotics against the pathogen *P. fusca*. The purification of such lipopeptide compounds corroborated the protective action against fungal pathogens of melon plants. Thus, the production of lipopeptides by different PGPR species has been widely corroborated as an important factor in the competition and colonization of the rhizosphere of various plants [79].

## 8. Motility and Chemotaxis

Chemotaxis is the ability to perceive a chemical stimulus and coordinate movement towards a stimulus with the help of cellular elements such as flagella or pili. Bacterial flagella and pili allow the motility of bacteria, including those that inhabit the rhizosphere, mainly because of the chemical attraction exhibited by root exudates [91]. Motility is a key trait for the colonization of the rhizosphere by various rhizospheric species, including bacterial species such as *P. fluorescens*. Some studies have shown that bacterial mutants with reduced motility are poor competitors and lack efficient adhesion to the host plant roots, whereas hypermobile and more competitive phenotypic variants are selected in the rhizosphere [92,93].

One of the first studies to demonstrate the importance of bacterial motility in the rhizosphere, as well as the colonization of the internal compartments of the plant, was published by Böhm et al. [94]. In this study, two mutant strains were constructed: an insertional mutant of *pilT* and a deletion mutant of *pilA*, the major structural component of the pilus structure in the endophytic bacterium *Azoarcus* sp. strain BH72, thus abolishing twitching motility. Results on rice root colonization using gnotobiotic cultures showed that the establishment of microcolonies on the root surface was strongly reduced in the *pilA* mutant, while the *pilT* mutant revealed a 50% reduction in root surface colonization. Interestingly, both mutants showed impaired endophytic colonization. Thus, in this system, type IV pili are important in the first critical step for successful colonization and adhesion to the plant host roots.

Recently, Fernández-Llamosas et al. [93] showed that motility and adhesion activities are relevant for the efficient colonization of rice roots by the endophyte *Azoarcus* sp. CIB. In this study, knockout mutants of the gene *fliM*, encoding the FliM protein that is involved in flagellum movement, the gene *pilX*, which along with other *pil* genes, encodes proteins that participate in the motility apparatus; and the gene *epsF* which is involved in exopolysaccharide synthesis/modification. The three of these mutants showed a significant deficiency in rice root colonization compared to the wild-type strain. Interestingly, another gene involved in the synthesis of cyclic di-guanosine monophosphate (c-di-GMP) was also found to be involved in the efficient colonization of rice plant roots by strain CIB. The compound c-di-GMP controls the synthesis of exopolysaccharides, adhesins, and biofilm formation in bacteria.

Bacterial volatiles have been widely associated with biocontrol and plant growth stimulating activities; however, little is known about their role in modulating the motility patterns of bacteria. For example, Martínez-Cámara et al. [95] demonstrated that the volatile dimethylhexadecylamine (DMHDA) produced by the rhizospheric bacterium *Arthrobacter agilis* UMCV2 affects bacterial growth and swarming motility of bacteria in a species-specific manner. When DMHDA was added to the medium, it modulated the swarming motility of *Bacillus* sp. ZAP018 and *P. fluorescens* UM270, but not *P. aeruginosa* PA01. As the UMCV2 strain is an inhabitant of the rhizosphere, it is possible that the DMDHA compound, in addition to promoting plant growth and antifungal action, regulates some essential activities for the rhizosphere colonization by bacteria, such as motility, giving it a competitive advantage in such an environment.

Previous studies have demonstrated the importance of motility, adhesion, and molecular regulators in efficient plant root colonization. Since in silico studies may contribute to a better understanding of these functions in the rhizosphere environment, the complete genome of the PGPR *P. fluorescens* UM270 was sequenced and compared with other rhizosphere strains of the same species (i.e., strains Pf0-1, A506, F113, SBW25, PICF-7, UK4, and UW4). In this way, several unique genes involved in rhizospheric colonization were identified along with 17 other coding regions related to bacterial motility and flagellar proteins. In particular, the biosynthetic proteins FlhABPQR, regulatory flagellar protein (FleQ), motor switch (FliN), and structural basal-body rod proteins (FlgCDFG) have been identified [96,97]. This analysis suggested that these functions were involved in rhizospheric colonization and competition, which is consistent with experimental evidence [86].

## 9. Evasion and Suppression of Plant Immune System

Unlike mammals, plants do not have mobile defense cells or an adaptive immune system. However, they have developed a rather sophisticated innate immune system, which can detect possible invading microorganisms through the use of transmembrane pattern recognition receptors (PRRs), which respond to microbe-associated molecular patterns (MAMPs), which are molecules shared by a wide range of microorganisms [98]. Beneficial microorganisms are also detected by the immune system of plants as a potential danger, as they possess immunogenic MAMPs that are very similar to those of pathogens [99].

Beneficial microorganisms have been shown to activate the immune system of plants during their initial contact with roots, as they are recognized by plant PRRs [62,100]. For example, the bacterium *P. simiae* WCS417 activated the immune response in *Arabidopsis* roots by detecting the expression of genes that respond to MAMPs. Similarly, *Bradyrhizobium japonicum* stimulated the expression of defense genes in the early stages of their interaction with root hair cells [101]. However, it has also been observed that beneficial microorganisms have the ability to circumvent or repress the defense system to achieve successful rhizospheric and endophytic colonization [99,102].

### 9.1. Evasion of Plant Immune System

According to Zboralski and Filion [61], evasion primarily consists of preventing the activation of the plant immune system. Some of the mechanisms used by various PGPR are described below. Flagellin is the most important structural protein that is part of the bacterial flagella and is essential for the mobility of these bacteria [103]. *Arabidopsis* FLS2 PRRs generally recognize this bacterial protein and bind to the immunogenic epitope fgl22, a conserved 22 amino acid sequence located at the N-terminus. Following this binding, plants turn on their immune systems [104].

However, the bacterium *Sinorhizobium meliloti* has important differences in this highly conserved sequence, which is sufficient to disable the immune activation of *Arabidopsis* FLS2 [105]. Later, Lopez-Gomez et al. [106] demonstrated that purified flagellin, from the symbiont bacterium *Mesorhizobium loti*, did not activate the immune response of *Lotus japonicus*, whereas the epitope fgl22 from *P. aeruginosa* induced the production of ethylene and the activation of defense-related genes, among other molecules, from the same plant. Similar evidence was reported by Trdá et al. [107], who observed that the grapevine FLS2 receptor differentially recognized the flg22 epitope of the PGPR *B. phytofirmans* from the pathogenic bacteria *P. aeruginosa* and *Xanthomonas campestris*, since grapevine showed a significantly reduced immune response with the *B. phytofirmans* compared to the responses observed with the epitope of the pathogenic bacteria.

One of the most important mechanisms in bacterial evasion of plant immunity is a decrease in flagella synthesis. For example, in *Pseudomonas* spp., flagella synthesis is regulated by cyclic-di-GMP, which mediates the transition between the planktonic and sessile lifestyle [108]. Thus, high levels of cyclic-di-GMP inhibit flagellin synthesis, which, in turn, prevents ROS synthesis in plants [109].

On the other hand, the ability of some PGPR to hide certain immunogenic MAMPs is another evasion mechanism of the plant immune system [100]. To prevent the activation of the immune system, *Pseudomonas* spp. can produce an alkaline protease known as AprA, which has the ability to degrade monomers of the flagellin protein [110]. Berendsen et al. [111] detected homologs of AprA in strains of *P. fluorescens*, i.e., WCS374 and WCS417, which are common inhabitants of the rhizosphere, suggesting that this mechanism is not exclusive to pathogens such as *P. syringae* and *P. aeruginosa* [112]. According to Yu et al. [100], the above-mentioned experiments suggest that plants select the members of their microbiome, through their PRRs, while some rhizosphere bacteria have developed mechanisms to evade the plant immune response mediated by PRRs, with the aim of positively associating with plants.

### 9.2. Suppression of Plant Immune System

Pathogenic microorganisms synthesize effector proteins that interfere with the signaling processes of plant cells, particularly the type III secretion system (T3SS), whose effector proteins can inhibit components of the plant immune system [113]. Similarly, beneficial microorganisms use a wide range of effector proteins that suppress the plant immune system, including the T3SS system, which is present in numerous beneficial bacterial genomes, including members of the *Rhizobium* and *Pseudomonas* genera [114,115].

In the case of *Pseudomonas* spp., it was shown that inoculation of *A. thaliana* with *P. simiae* WCS417 suppressed more than half of the transcriptional responses activated by MAMPs. The authors of this work suggest that this may allow the establishment of a beneficial relationship between the bacteria and the roots of plants [100]. Thus far, these results suggest that the beneficial microorganisms in the rhizosphere also interfere with the signaling of the plant immune system through the production of effector molecules that suppress it. However, given the very few studies on this subject, much remains to be investigated in this regard. Table 1 summarizes the different mechanisms of rhizosphere colonization in PGPR so far reviewed.

## 10. Conclusions and Future Perspectives

In this review, some important functions that bacteria exhibit in order to compete, colonize, and establish themselves in the rhizosphere of plants have been discussed. The use of competitive PGPR in open field experiments has shown to be an important action to reduce or eliminate the application of chemicals in agricultural production [116]. The mechanisms used by these bacteria include recognition of chemical signals and nutrient uptake, antioxidant activities, biofilm production, and motility [61,117,118]. In addition, the production of antimicrobial compounds helps PGPR compete with other microorganisms that cohabit the rhizosphere [13,78,119]. However, several of these mechanisms are also employed by plant pathogens; thus, isolating and selecting the most competitive PGPR is essential to obtain good results in the field. Additionally, these strains must be able to stimulate plant growth once they have established themselves in the rhizosphere. Some rhizosphere strains also interact with the plant, penetrate the root tissues and colonize other parts of the plant, such as stems, leaves, and fruits, from where they could also exert stimulatory or protective functions against non-rhizosphere pathogens [120]. 

The detailed mechanisms used by PGPR in the rhizosphere are still being elaborated. Some studies have shown that certain activities such as biofilm production are important for establishing and attaching to plant roots, while other studies propose that such functions play a role in colonization [61,68,92,93,121]. If a function is important, it is essential to observe that the experimental conditions under it are operative; this is because there are many biotic and abiotic factors that modulate the beneficial effects of PGPR.

Recently, massive sequencing techniques, including metagenomic and diversity studies of ribosomal genes or ITSs, or denaturing gradient gel electrophoresis profiles, have shown the impact of PGPR inoculation in the rhizosphere of plants on the resident microbiota [122,123]. Studies using these techniques can also reveal whether colonizing PGPR can indirectly have a beneficial effect on the plant, either by increasing growth-stimulating taxa or antagonizing potential pathogens. However, it is essential to isolate and characterize such strains and confirm their beneficial activities to avoid undesired results in crops [13,124].

Finally, it would be beneficial to establish new screening methods for important functions in PGPR to colonize the rhizosphere, as only a few methodologies have been proposed [125]. In addition, it is important to define the specific functions of some mechanisms for promoting plant growth in rhizosphere colonization; for example, some volatiles have shown antibiotic action under laboratory conditions; however, a significant role in the rhizosphere has been observed in a few studies [126]. Filling in all the gaps that exist in the knowledge of mechanisms of rhizosphere colonization by PGPR would facilitate the selection of superior strains as bioinoculants in the field.

## Figures and Tables

**Figure 1 biology-10-00475-f001:**
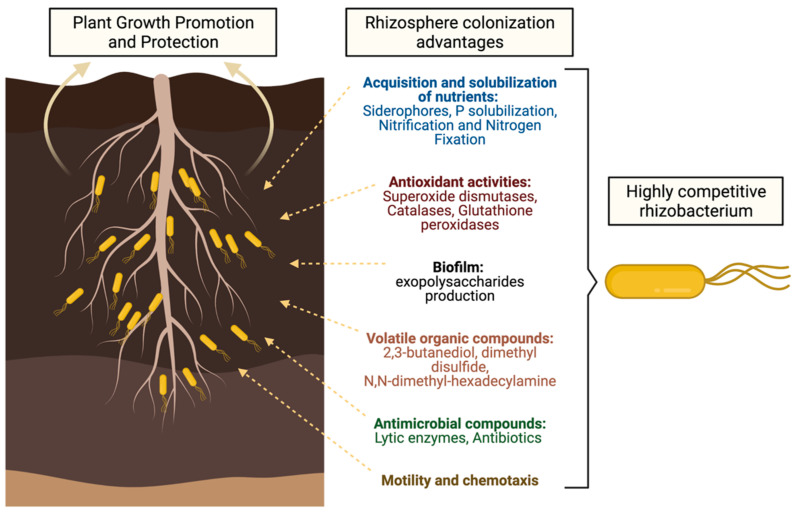
Schematic description of the main mechanisms used by rhizobacteria to competitively colonize the rhizosphere of host plants. See below for more details.

**Table 1 biology-10-00475-t001:** Recent works detailing the diverse mechanisms of PGPR to efficiently colonize rhizospheres of plants.

Mechanisms	Main Benefits of Mechanisms	References
Root exudates	Chemo-attractants of beneficial bacteria in the rhizosphere.	[18,30]
Acquisition and nutrients solubilization: Siderophores and acquisition of iron Phosphate solubilization Nitrification and nitrogen fixation	Siderophores are ferric ion specific chelators, and they possess antimicrobial properties. Phosphate solubilizing microorganisms mediate bioavailable soil P to plants. Benefit the host plant mainly by fixing atmospheric nitrogen.	[4,14,26,36,37,38,45,48,51]
Antioxidant activities	PGPRs maintain high antioxidant enzyme activity under stress conditions.	[56,57]
Biofilm production	The nitrogenase activity, IAA production, phosphate solubilization, siderophore production, ammonia production, and higher resistance to adverse environmental are higher in PGPR biofilm than the planktonic cells.	[63,64]
Volatile organic compounds	Increase the biosynthesis of secondary metabolites, improve the antioxidant status in some plants grown under salt stress, and inhibit pathogenic fungi such as *B. cinerea.*	[95]
Antimicrobial compounds production: Lytic enzymes Antibiotics	The extracellular hydrolytic enzymes degrade cell wall components of plant pathogenic microbes. The main mechanism by which PGPRs biocontrol plant pathogens is the antibiotics production.	[83,88,93,94,95]
Motility and chemotaxis	Successful colonization will only be achieved if preceded by the detection (chemotaxis) of root exudates, and movement (motility) of the microorganisms towards the plant roots.	
Evasion and suppression of plant immune system: Evasion of plant immune system Suppression of plant immune system	Evasion consists, primarily, of preventing the activation of the plant immune system. Suppression refers to bypassing the plant immune system, through the use of effector proteins.	[99,100]

## Data Availability

Not applicable.

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
