# Peer review of "Rhizosphere Colonization Determinants by Plant Growth-Promoting Rhizobacteria (PGPR)"

_biology, 2021, doi:10.3390/biology10060475_

Round 1
Reviewer 1 Report
General comments
In general, this review article has important topic but I have some minor comments.
Detailed comments
Abstract
This section is poorly written and missing the direct aim of this review article (please rewrite)
7.1 Lytic Enzymes
Line 346-358
what the author mean by this information. please clarify
Findings
Although, I think the data provided was short, it is scientifically sound conclusions and future perspectives:
Line 494 please change Here to.. In this review articles
Author Response
In general, this review article has important topic but I have some minor comments.
Response: Thank you.
Detailed comments
Abstract
This section is poorly written and missing the direct aim of this review article (please rewrite)
Response: Thank you for your comment, we included the Abstract was modified and added the following sentence: ¨Therefore, in this work, it is analyzed in detail the different mechanisms of rhizosphere colonization in PGPR, as well as discussing the advantages of having highly competitive strains for a better open-field performance¨
7.1 Lytic Enzymes
Line 346-358
what the author mean by this information. please clarify
Response: modified as suggested, thanks.
Findings
Although, I think the data provided was short, it is scientifically sound conclusions and future perspectives:
Line 494 please change Here to.. In this review articles
Response: modified as suggested, thanks.
Reviewer 2 Report
several words need btter phrasing and acronyms must be eplained in clear see attach file

Author Response
Thank you so much for your comments and suggestions, we greatly acknowledge them!
About the title, we considered the alternative words but preferred to stick with determinants.
All the rest of the comments were considered and modifications were done accordingly. Please see attached R1 file with highlighted changes.
Reviewer 3 Report
The revision “Rhizosphere colonization determinants by plant growth-promoting rhizobacteria (PGPR)” written by Santoyo et al. presents relevant and important subject and the paper is very well written.
I have only one suggestion to the authors: The authors could address in the review some aspects related to the compatibility of pesticides with microbial inoculants, since it is a determinant issue in the success of PGPR colonization.
Recently, a great revision regarding “The Challenge of Combining High Yields with Environmentally Friendly Bioproducts: A Review on the Compatibility of Pesticides with Microbial Inoculants” was published in Agronomy – Santos et al. (2021). https://doi.org/10.3390/agronomy11050870
Author Response
Thank you for your comments and suggestion. We included the reference with the following paragraph in Conclusions and future perspectives.
¨The use of competitive PGPR in open field experiments has shown to be an important action to reduce or eliminate the application of chemicals in agricultural production [118].¨